# Untangling Synergistic Effects of Intersecting Social Identities with Partial Information Decomposition

**DOI:** 10.3390/e24101387

**Published:** 2022-09-28

**Authors:** Thomas F. Varley, Patrick Kaminski

**Affiliations:** 1School of Informatics, Computing, and Engineering, Indiana University, Bloomington, IN 47405, USA; 2Department of Psychology & Brain Sciences, Indiana University, Bloomington, IN 47405, USA; 3Department of Sociology, Indiana University, Bloomington, IN 47405, USA

**Keywords:** partial information decomposition, synergy, sociology, intersectionality, higher-order interactions

## Abstract

The theory of intersectionality proposes that an individual’s experience of society has aspects that are irreducible to the sum of one’s various identities considered individually, but are “greater than the sum of their parts”. In recent years, this framework has become a frequent topic of discussion both in social sciences and among popular movements for social justice. In this work, we show that the effects of intersectional identities can be statistically observed in empirical data using information theory, particularly the *partial information decomposition* framework. We show that, when considering the predictive relationship between various identity categories such as race and sex, on outcomes such as income, health and wellness, robust statistical synergies appear. These synergies show that there are joint-effects of identities on outcomes that are irreducible to any identity considered individually and only appear when specific categories are considered together (for example, there is a large, synergistic effect of race and sex considered jointly on income irreducible to either race or sex). Furthermore, these synergies are robust over time, remaining largely constant year-to-year. We then show using synthetic data that the most widely used method of assessing intersectionalities in data (linear regression with multiplicative interaction coefficients) fails to disambiguate between truly synergistic, greater-than-the-sum-of-their-parts interactions, and redundant interactions. We explore the significance of these two distinct types of interactions in the context of making inferences about intersectional relationships in data and the importance of being able to reliably differentiate the two. Finally, we conclude that information theory, as a model-free framework sensitive to nonlinearities and synergies in data, is a natural method by which to explore the space of higher-order social dynamics.

## 1. Introduction

“Intersectionality” refers to the idea that an individual’s experience of social privilege or oppression is a function of how all the various identities that a given person can hold (such as race, sex, class, ability, etc.) intersect, and that the result of these intersecting identities is not directly decomposable into the sum of all identities considered individually. Intersectionality as a framework was first articulated by Kimberle Crenshaw [1,2], who discussed how Black women in particular face distinct forms of marginalization in the context of the American justice system. She argued that pre-existing theoretical frameworks of both feminist and anti-racist scholarship erased the specific experiences of Black women, instead operating under a framework where *“All the Women Are White; All the Blacks Are Men”* [1]. The particular intersection of Blackness and race has been called *misogynoir* by Moya Bailey [3] to articulate its status as a unique, irreducible social experience.

From its initial focus on Black feminism and the unique experience of Black women, the field of Intersectionality Studies has expanded [4] and now many different axes of identity are routinely studied, including sexuality [5,6], class [7], disability [8,9], and/or immigration status and citizenship [10,11,12,13]. Outside of academia, intersectionality has gained prominence in popular culture, particularly around issues related to the Black Lives Matter movement, trans and gender-nonconforming rights, and the issue of immigration from South and Central America into the United States. As a consequence of the growing public discussion and activism, there has been increasing interest in how intersectional frameworks could be deliberately incorporated into public policy decisions [14,15,16]. As theories of intersectionality leave the Academy and begin to influence more mainstream decision-making, it is imperative that the field develops analytic tools that allow for the discussion of intersecting identities in the context of empirical data. Understanding the causal and material impacts of policy given intersectional identities requires the development of robust empirical methodologies that can be used to inform and identify interventions.

Previous researchers have highlighted the difficulties in addressing the issue of synergistic relationships between identities in empirical data: for example Bowleg concisely summarizes the issue as “*Black + Lesbian + Woman*≠*Black Lesbian Woman*”, and discusses the problems with assuming additive relationships, although Boweleg is primarily concerned with analysing qualitative ethnographic data and does not engage with the mathematical issues around super-additive relationships in numerical or categorical data [17]. There has been discussion of how intersectionality can be accounted for by quantitative methods, for example Scott and Siltanen [18] discuss multi-level linear regressions informed by context, and Rouhani provides a gold-standard primer detailing how to distinguish between additive and multiplicative effects using linear regressions [19]. Generally, the standard practice is identifying intersectional effects with the multiplicative interaction term in linear regression and comparing it to the main effect using estimators of prediction error like Akaike’s Information Criteria.

While powerful, these methods have a number of fundamental limitations that can complicate the analysis of complex data. The reliance on parametric models, goodness-of-fit tests, and arbitrary criteria such as the α-level of statistical significance can limit the kinds of relationships the analysis is sensitive to. For example, the reviewed literature makes overwhelming use of linear (Gaussian) assumptions in assessing main and interaction effects, and does not account for non-linear relationships between interacting variables. Another significant concern is that, while linear regressions can compare main and interaction effects, and in doing so account for interactions between identities, it fails to capture the synergistic core of intersectionality: no distinction is made between different kinds of interactions, such as redundant vs. synergistic dependencies. For example, a Black woman plausibly experiences: generic anti-Black racism (independent of sex), generic misogyny (independent of race), and intersectional misogynoire specific to her identities as a Black woman. Furthermore, there may also be “redundant” effects: a cost shared by Black people and women by virtue of being a minority of any type and not specific to either identity. Linear regression models fail to decompose this constellation of effects, as they have no way to rigorously account for, or even acknowledge, differences between redundant and synergistic relationships. This inability has implications for our ability to model the social dynamics that give rise to systemic, higher-order inequalities, as different kinds of generative dynamics may give rise to redundant or synergistic dependencies. For example, redundant dependencies are often reflective of a common-driver effects [20] which are often conditioned out, while synergies represent irreducible, higher-order interactions that cannot be simplified and posses their own structure reflecting multivariate dependencies in the data. To address these issues, we turn to an alternative statistical framework for analysing data: information theory.

Information theory represents an appealing statistical framework with which to tackle the question of intersectionality in quantitative data: in contrast to standard linear regression models, information theory usually does not require making any assumptions about the distribution of the data (normal, Poisson, etc.), making it sensitive to non-linear relationships in data [21]. It is well-known that Gaussian models can fail when the underlying generative dynamic is strongly non-normal, such as in heavy-tailed distributions (for example, the distribution of wealth in the United States) [22,23], temporally extended human dynamics, such as epidemic spreading [24] and these nonlinearities can lead to erroneous conclusions and incorrect models [25]. Finally, for researchers who do wish to leverage the power of linear models, or work with continuous data, closed-form Gaussian estimators of all major information-theoretic relationships exist: for example, the Pearson correlation is a function of the mutual information between Gaussian variables [21], the partial correlation maps to the conditional mutual information [26], and the Granger Causality is a special case of the more general transfer entropy [27]. There also exist non-parametric, continuous estimators based on K-nearest neighbor relationships [28] for non-linear analysis of real-valued (continuous) data (also see [29], Appendix A).

Crucially, information theory is well-equipped to handle the problem of decomposing multivariate relationships in data into synergistic (intersectional) and redundant components using a framework known as partial information decomposition (PID) [30,31] (see Section 2.2 for details), and has been applied in a variety of fields, including interpretable machine learning [32], medical imaging [33], biological neural networks [34,35], ecology [36], evolution [37], as well as to philosophical questions such as the nature of “emergence” [38,39,40] and consciousness [41]. This interdisciplinary group of results suggests that synergistic relationships “greater than the sum of their parts” are ubiquitous in both natural and human-made systems, so it is natural to hypothesize that they may also exist in social systems.

In this paper, we propose a conceptual link between synergisitc, “greater than the sum of their parts” information with intersectionality in sociological data. By applying the partial information decomposition analysis to US census data, we can test for higher-order interactions between multiple identities (race, sex, etc.) on outcomes like income or health status. We hypothesized that statistical synergies indicating the presence of intersectional inequalities (i.e., costs associated with being both Black and female or advantages associated with being both White and male) should be observable in population-level demographic and life-outcome data using information theoretic analysis, and that information theory will out-perform linear regression in discriminating between redundant and synergistic effects. Furthermore, we go on to compare the information-theoretic analysis with standard, multivariate linear models to show that the linear approaches are not equipped to differentiate between different kinds of multivariate dependence.

## 2. Methods

### 2.1. Basic Information Theory

Information theory is a mathematical framework that describes how different interacting entities inform on and constrain each-other’s behaviour [21]. Originally developed in the context of theories of communications [42], information theory has become an indispensable tool for the analysis of complex systems [20,43].

#### 2.1.1. Information Theory & Inference

To build an intuition about how information theory can help us understand interactions between different identities, it is helpful to consider simple examples. Imagine selecting a person at random and, knowing nothing about their sex, race, or any other demographic considerations, trying to predict their income. In the absence of any contextual information, the maximally likely income is the expected value of the whole income distribution (in this case $43,308.13 USD, see Section 2.4), although there is going to be variability within the population which makes us uncertain about the accuracy of our estimate (the entropy, see below). Now, suppose that we learn the sex of our randomly selected person (male or female). Knowing this new piece of information will *reduce our uncertainty* about their likely income by ruling out everyone inconsistent with the target sex. In a fundamental way, this is what it means to say that sex discloses information about income. We can walk through a similar example with the category race. Suppose we pick a new person at random, and learn their race (White or Black). Similarly, our initial uncertainty about income is reduced, because race and income are correlated. Finally, suppose we learn both the race *and* the sex of our randomly selected person: knowing the joint state of both predictors will reduce our uncertainty about income the most. If we learn *more* information from knowing both demographic variables together than we learn from the sum of the two predictors considered independently, then there is some information in the *whole* that is greater than the sum of it’s parts—what we call the *intersectional synergy*. This is a distinct concept from the multiplicative interaction term more commonly used in regression analysis as it does not necessarily assume a particular kind of interaction (multiplicative), but instead is based on the information that is *only* accessible when considering both identities jointly. As we will see, the information-theoretic perspective allows us to explore other dependencies, such as the difference between learning race *and* sex compared to learning race *or* sex (which correspond to synergy and redundancy, respectively). These represent two different kinds of interaction that can be disentangled, while the regression only gives us one kind of interaction.

#### 2.1.2. Entropy & Mutual Information

The core object of study in information theory is the *entropy*, which quantifies how uncertain we, as observers, are about the state of a variable we are observing. Inference, then, is typically understood as the process of minimizing entropy by understanding how information about the variable in question is disclosed by both it’s own statistics, and related variables. Given a variable *X* which can take on different states *x* drawn from a support set X according to the probability distribution $P_{X}\left(x\right)$, the entropy of the variable is given by:(1)$$H\left(X\right)=-\sum_{x\in X}P_{X}\left(x\right)\log\left(P_{X}\left(x\right)\right).$$

If we are observing more than one variable, the joint entropy is an easy generalization: for a pair of variables {X,Y}, the joint entropy is given by:(2)$$H\left(X,Y\right)=-\sum_{x\in X}\sum_{y\in Y}P_{X,Y}\left(x,y\right)\log\left(P_{X,Y}\left(x,y\right)\right)$$

Which quantities the total uncertainty about the state of both variables simultaneously. From the joint and marginal entropies we can calculate the conditional entropy:(3)H(X|Y)=H(X,Y)−H(Y).

This conditional entropy tells us how much uncertainty about the state of *X* remains after we have “accounted for” the state of *Y*. It is important to note that H(X|Y)≤H(X): information about the state of *Y* can only ever reduce our uncertainty about the state of *X* or, if *X* and *Y* are independent, provide no insight.

So far, all the measures that we have discussed have been measures of uncertainty, not measures of information. To quantify information itself, we introduce the mutual information:(4)I(X;Y)=H(X)−H(X|Y).

For a visual intuitive aid, see Figure 1.

It is worth unpacking this to build intuition: we begin with our uncertainty about the state of *Y*, given by H(Y). From this, we subtract our remaining uncertainty about *Y* after the state of *X* has been taken into account (H(Y|X)). The difference of these two quantities is our uncertainty about *Y* that is *resolved* by knowledge of *X*: the amount of information *X* provides about *Y*. It is important to note that the mutual information is symmetric: I(X;Y)=I(Y;X). As with the joint entropy, it is possible to calculate the mutual information between a set of variables and a single target. Consider the case of trying to predict the state of a variable *Y* based on two predictor variables $X_{1}$ and $X_{2}$. We can calculate $I(X_{1},X_{2};Y)$ the same way as above, only treating the joint states of $X_{1}$ and $X_{2}$ as a single macro-variable.

It is crucial that the joint mutual information is **not** always equivalent to the sum of the individual marginal mutual informations:(5)$$I\left(X_{1},X_{2};Y\right)\neq I\left(X_{1};Y\right)+I\left(X_{2};Y\right).$$

The joint mutual information can be either greater, or less than, the sum of the marginal mutual informations, depending on how correlated information is distributed across the two predictor variables. If $I\left(X_{1},X_{2};Y\right)<I\left(X_{1};Y\right)+I\left(X_{2};Y\right)$ then the total is less than the sum of their parts and consequently $X_{1}$ and $X_{2}$ must share some *redundant* information. Consider Venn diagram in Figure 2: if the area of two overlapping circles is less than the sum of the areas of both circles considered independently, then there is redundant area shared between them. Conversely if $I\left(X_{1},X_{2};Y\right)>I\left(X_{1},Y\right)+I\left(X_{2},Y\right)$, then the whole is *greater* than the sum of the parts: there is information about *Y* in the joint state of $X_{1},X_{2}$ that cannot be extracted from either variable considered independently. In the case of two predictors and a single target variable, the difference between the whole and the sum of the parts (sometimes called the “Whole Minus Sum” integration [44]), is equivalent to a number of other information-theoretic measures, such as the co-information (a trivariate generalization of mutual information) and related to a recently proposed measure of redundancy/synergy bias: the O-information [45]. In the general case, it has also been introduced under the name Redundancy/Synergy Index [46].

Finally, it is entirely possible that a pair of predictor variables contains *both* redundant and synergistic information: while the difference between the joint and the sum of the marginal mutual informations is a decent heuristic to test if there is any synergy at all in the triad, it does not tell us *how much* redundant or synergistic information is present, nor what proportion of the total information is accounted for by redundant or synergistic information. For this we need more advanced mathematical machinery.

### 2.2. Partial Information Decomposition

Given a set of predictor variables influencing a single predicted variable, the partial information decomposition framework [30,31] provides the tools necessary to decompose the joint mutual information into redundant, synergistic, and unique “types”. The resulting decomposition is given by:(6)$$\begin{array}{ccc} I(X_{1},X_{2};Y) & = & Red\left(X_{1},X_{2};Y\right)+Unq\left(X_{1};Y/X_{2}\right)+Unq\left(X_{2};Y/X_{1}\right)+Syn\left(X_{1},X_{2};Y\right). \end{array}$$
where $Red(X_{1},X_{2},Y)$ quantifies the information about *Y* that can be resolved by observing $X_{1}$ OR $X_{2}$, $Unq(X_{1};Y/X_{2})$ quantifies the amount of information about *Y* is uniquely disclosed by $X_{1}$ (in the context of $X_{2}$) and vice versa. Finally, $Syn(X_{1},X_{2};Y)$ quantifies the information disclosed only by the joint state of $X_{1}$ AND $X_{2}$, and no simpler combination of variables. We can decompose the marginal mutual informations in the same fashion:(7)$$\begin{array}{ccc} I(X_{1};Y) & = & Red\left(X_{1},X_{2};Y\right)+Unq\left(X_{1};Y/X_{2}\right), \end{array}$$(8)$$\begin{array}{ccc} I(X_{2};Y) & = & Red\left(X_{1},X_{2};Y\right)+Unq\left(X_{2};Y/X_{1}\right). \end{array}$$

The result is a system of three equations, three known quantities (the joint and marginal mutual informations), and four unknown variables (*Red, Unq${}_{1}$, Unq${}_{2}$, Syn*, referred to as the “atomic” components of the mutual information). If it is possible to identify any one partial information term, then the remainder are trivial. The synergy term can also be written as the difference between the joint mutual information and all the simpler combination of sources:(9)$$Syn\left(X_{1},X_{2};Y\right)=I\left(X_{1},X_{2};Y\right)-Red\left(X_{1},X_{2};Y\right)-Unq\left(X_{1};Y/X_{2}\right)-Unq\left(X_{2};Y/X_{1}\right).$$

In this perspective, we can see that synergy is the information about the target that is accessible when observing the whole (both sources) and none of the parts considered independently. In Figure 2, it is equivalent to the violet area of the oval. This definition of synergy as the difference between the joint state of both predictors (AND) and the redundant information (OR) provides a useful framework with which the PID can be connected to other relevant analyses based on logical conjunctions and disjunctions (see Section 3.1).

### 2.3. Choosing a Partial-Information Function

Unfortunately, classical Shannon information theory does not provide a function that calculates any of these, and neither does the PID framework itself. Much work has gone into developing such functions, and as yet, there is no consensus within the field of information theory as to the “gold standard”. To ensure the robustness of the concept, we analysed the data using two different functions to ensure that the distributions of redundancy and synergy remained consistent despite different technical starting points. The first was the original measure proposed by Williams and Beer [30] called $I_{min}$, which defines the redundant information shared by a set of sources about a target as:(10)$$I_{min}\left(X_{1},X_{2},\ldots ,X_{n};Y\right)=\sum_{y\in Y}P_{Y}\left(Y=y\right)\min_{i}I\left(X_{i};Y=y\right).$$

This measure (also called the “specific information”) gives the minimum amount of information that the set of sources discloses about the target. To ensure that our results were robust to the particular free parameters inherent in the PID framework, we replicated our results using the measure of $I_{BROJA}$ proposed by Bertschinger et al. [47,48]. The $I_{BROJA}$ method starts with the unique information, rather than the redundant information; defining it as the minimum conditional mutual information possible while holding the marginals constant. To extract the unique information $Unq(X_{1};Y/X_{1})$, Bertschiner et al., begins by defining the set of all joint distributions of $X_{1}$, $X_{2}$, and *Y* such that the marginals are equivalent to the empirical distribution *P*:(11)$$\Delta =\{Q|P_{X,Y}\left(x_{i},y\right)=Q_{X,Y}\left(x_{i},y\right)\forall i\}.$$

The unique $I_{BROJA}$ for a given source is defined as:(12)$$I_{BROJA}\left(X_{1};Y/X_{2}\right)=\min_{Q\in \Delta }I^{Q}\left(X_{1};Y|X_{2}\right).$$

Where $I^{Q}\left(\cdot ;\cdot |\cdot \right)$ indicates that the mutual information being calculated with respect to the probability distribution $Q(X_{1},X_{2},Y)$. Once the two unique informations have been calculated, the rest of the terms can be solved with basic algebra. Both $I_{min}$ and $I_{BROJA}$ provide strictly positive values for all partial information atoms, allowing for a complete decomposition of the joint mutual information terms. As mentioned above, controversies and discussions remain about the “best” approach to defining partial information functions: both $I_{min}$ and $I_{BROJA}$ have their own distinct limitations and particular behaviours. $I_{min}$ has been criticised for behaving in “counter-intuitive” ways in some edge cases [49], although $I_{min}$ remains by far the most widely-used redundancy function in analysis (see [32,34,39,50] for a non-exhaustive list). $I_{BROJA}$ was developed in part in response to the limitations of $I_{min}$, although it is restricted to the triadic case of two predictors and a single target and cannot be readily generalized to more complex structures, unlike $I_{min}$. In the main body of the paper we report the results of the $I_{min}$ analysis, and the results of $I_{BROJA}$ (where applicable) are available in the Appendix A.

### 2.4. Data & Pre-Processing

For this project, we used a decade of data (2011–2020) from the Annual Social and Economic (ASEC) Supplement (available here: https://www.census.gov/data/datasets/time-series/demo/cps/cps-asec.2020.html, accessed on 9 September 2021), which provides individual-level micro-data on demographic and economic indicators in the United States. We excluded individuals younger than 25 years of age, or older than 65 years of age as they were less likely to be working full-time (children, students, or retired), individuals not native-born US Citizens, and we only analyzed those who self-identified as ”White only” or “Black only” to limit the confounds associated with multi-racial identities. Finally, we only considered individuals who were working in full-time employment since we were looking explicitly at the effects of identities like race or sex on income and the possibility of different levels of employment would represent a possible confound.

To limit the size of the joint probability space, we coarse-grained the income distribution: individuals making $0.00–$27,499 were placed into a Low Income group, individuals making $27,500–$52,499 where placed into a Lower-Middle Income group, individuals making $52,500–$77,499 where placed into Upper Middle Income group, and finally those making more than $77,500 were placed into an High Income group, although we found that our results were generally robust to the exact choice of cutoffs or the number of categories up to a point at which point the joint probability distribution was too sparsely sampled. When considering health outcomes, we did not coarse grain the data, which was reported on a 5-item Likert scale.

### 2.5. Analysis Pipeline

Performing the PID analysis requires constructing the entire joint-probability distribution for all variables of interest. For example, when untangling the synergistic effects of sex and race on income, it is necessary to compute:$$\begin{array}{cc} & P(Race=Black\wedge Sex=Male\wedge Income=Low),\\ & P(Race=White\wedge Sex=Male\wedge Income=Low),\\ & P(Race=Black\wedge Sex=Female\wedge Income=Low),\\ & P(Race=White\wedge Sex=Female\wedge Income=Low),\\ & \ldots \\ & P(Race=White\wedge Sex=Female\wedge Income=High), \end{array}$$
where ∧ is the logical-AND operator.

To accurately estimate these probabilities, it is crucial to ensure that the distribution of joint states of the predictor variables (in this case, race and sex) are evenly distributed: there should be an equal number of Black women, White women, Black men, and White men. If, for example, White men are over-represented and have higher-than-average incomes, which is the case in the full dataset, then that will incorrectly skew the influence of White male data points on the joint distribution and expected moments. To address this issue, we created 1000 sub-sampled distributions while keeping the number of joint-states of predictive variables the same (i.e., the same number of Black women, White women, Black men, and White men). We then calculated the mixture distribution of all 1000 sub-sampled distributions and used that to perform the analysis. By resampling the data to enforce a maximum-entropy distribution, we are controlling for the problem of uneven sampling, such as oversampling done by the census-takers to ensure sufficient coverage.

While the resampling addresses some sampling biases (such as uneven numbers of one particular demographic group skewing estimates), there is also an issue of bias in the inference of the probability distributions and subsequent entropies themselves. Here we use the “naive” or maximum-likelihood estimator, where P(X=x) is estimated by the number of times *x* appears in the dataset, normalized by the total number of samples. The MLE estimator is known to have a persistent bias for small datasets (for review, see [20]), however given the very large number of samples in the data (on average 70718.9 ± 8435.99 respondents in a given year), the effect of these biases is negligible. Furthermore, while bias corrections for entropy and mutual information are well-explored, in the context of the PID bias corrections are almost entirely unexplored, and so we leave this issue for future research.

By forcing a “flat” probability distribution on the space of joint identities, this can be understood as being a “causal” analysis in the sense of Pearl’s *do*-calculus model of causality [51,52] and Woodward’s interventionist causal framework [53]. Specifically, we are leveraging the measure of “effective information” (first introduced by Tononi and Sporns [54]) which quantifies all of the causally-relevant relationships between two variables:(13)$$EI\left(X\to Y\right)=I\left(X^{H_{max}};Y\right).$$
where $X^{H_{max}}$ indicates that the probability distribution of the states of the predictor variable *X* are maximally entropic (in this case, uniform). The maximum entropy distribution is considered to be “causal” because it controls for any biases that might be introduced from the empirical distribution and preserves only the core “effective” relationships in the data. The consequence is that, by strictly forcing our distribution of identities to be uniform (i.e., *P(Race = Black ∧ Sex = Female) = P(Race = White ∧ Sex = Male)*), we are decomposing the effective information shared between identity and outcomes (as recorded in this data).

We compared three different sets of variables: *I(Race, Sex; Income), I(Sex, Income; Health)* and *I(Race, Sex, Income; Health)*, which represent a sampling of the interactions of three major identity groups (with income standing in as a proxy for class). We began by comparing the difference between the information in the “whole” (both identities considered together) to the sum of the “parts” (the information each individual identity added together), which serves as a useful heuristic for whether the relationship is dominated by intersectional synergy or not. Subsequently we did the full partial information decomposition to extract the relevant partial information-sharing modes. To compare between datasets, each value was normalized by the joint mutual information, giving the proportion of the total information (i.e., of all the information that race and sex provide about class, what proportion of that information is synergistic, redundant, unique, etc.).

Finally, we show that redundant and synergistic information make distinctly different predictions about the effects of identity, on outcomes, and discuss how these distinct modes of interaction can form a basis for modeling cross-identity solidarity and exclusive intersectional experiences, respectively.

### 2.6. Data & Code Availability

All data can be downloaded from the US Census Bureau website ( https://www.census.gov/data/datasets/time-series/demo/cps/cps-asec.2020.html, accessed on 9 September 2021). All scripts necessary to recreate the analysis and figures are included as Appendix A. Data cleaning was done using the Pandas and Numpy packages and all information theoretic analysis was carried out in Python, using the *Discrete Information Theory* toolbox by James et al. [55]. Regression analysis was done using the Statsmodels package in Python [56].

## 3. Results

We began by examining the difference between the joint mutual information and the sum of the marginal entropies for each of our three relationships of interest across the whole decade (Figure 3). This is a practical heuristic sometimes called the “Whole Minus Sum” method [44,48] that quantifies the overall degree of redundancy or synergy-dominance in data. We found that, for all three relationships and across all ten years, the joint entropy was greater than the sum of the marginals, indicating that the relationship between various intersecting identities and target outcomes are consistently “greater than the sum of their parts,” or in the context of PID, “synergy-dominated”. The Whole Minus Sum heuristic provides a single, scalar measure that describes the interaction between multiple variables. In that respect, it is similar to standard linear regression approaches. If we compute a standard linear regression between race and sex on income, we see a single interaction effect (all coefficients significant at $p<10^{-10}$).

While this tells us that there is some significant interaction between race and sex with respect to income, it tells us very little about the structure of that interaction. The coefficient is a single, scalar value and consequently cannot distinguish between the different kinds of interactions (redundant or synergistic). To determine the exact distributions of redundant, unique, and synergistic information, we then computed the full PIDs using the $I_{min}$ redundancy function (for visualization of the two triadic interactions, see Figure 4, for visualization of the results with $I_{BROJA}$ see Appendix A). We found that the relationship between race and sex on income had a high degree of both synergy and redundancy, with only limited unique information disclosed by only race or sex. On average, 51±1.3% of the information disclosed by race and sex about income was synergistic in nature, while 42±0.8% of the information was redundantly shared between the two predictor variables. The two unique components together disclosed only 7±2% bit total. This decomposition into four different information-sharing modes shows that the general interaction revealed by the multiplicative interaction term in Table 1 is actually comprised of two different types of interaction, both present in roughly equal proportions, while the particular implications of redundant and synergistic interactions in this context remains a question for further study, these results highlight how information decomposition-based approaches can reveal structure in data that linear regression is insensitive to.

We can also see that the particular structure of different interactions can vary widely depending on the particular variables being analyzed. In contrast to the interaction between race, sex, and income (where redundant and synergisitc interactions dominated in roughly equal measure), the information that race and income provided about health status had a dramatically different structure: only 15.2±2.2% of information were synergistic, and only 13.27±1.8% was redundantly shared. Income uniquely disclosed 71.13±3.88% of the information about health, with only 0.4±0.82% being uniquely disclosed by race. Despite the clear differences in the breakdown of partial information, the joint mutual informations were very similar: *I(Race, Sex; Income)* = 0.057 bit and *I(Race, Income; Health)* = 0.06 bit—it is only when breaking down the information into it’s atomic components that we see meaningful differences. When considering the three-way effect of race, sex, and income on health, we found that 12.1±1.9% of the total information was some form of redundant (*Red(Race, Sex, Income), Red(Race, Sex), Red(Race, Income), Red(Race, Sex)*), while another 15.48±2.31% was some form of synergistic. Unique information account for 68.88±3.76% of the total. Note that these do not sum up to 100% since there is a small amount of information distributed over more exotic and hard-to-interpret dependencies.

When considering all of the years individually, it is clear that the informational relationships are stable over time (as evidenced by the comparatively low standard deviation terms). There does not appear to be a substantive increase, or decrease in how the various identities interact, suggesting that the intersectional relationships are consistent over multiple years. When comparing the results of the $I_{min}$ measure with the results of using the $I_{BROJA}$ measure, we found that the patterns were overwhelmingly similar: *I(Race, Sex; Income)* was comprised largely of equal parts redundant and synergistic information, while *I(Race, Income; Health)* decomposed into primarily information uniquely disclosed by income. Since $I_{BROJA}$ cannot accomodate more than two predictors, there was no way to calculate the joint-effects of race, sex, and income together on health. See Appendix A for visualization.

### 3.1. Untangling Redundant & Synergistic Information

The partial information decomposition framework describes two distinct ways that different identities can interact they can both individually communicate the same information (redundancy) or they can jointly communicate information that is not not disclosed by any simpler combination of sources (synergy), which we identify with “intersectionality”. Following [31] and [57], we can relate the redundant and synergistic information to the logical disjunction (OR) and logical conjunction (AND), respectively. Consider the redundant information between two variables: it is information that is disclosed by *either* variable. An observer could choose one of the two variables at random, observe it and only it, and learn the same information as if they had chosen the other variable.

When considering the information redundantly present in both race and sex about income, we ask: how would we revise our estimate of a given person’s income if we knew that they were Black *or* a woman? Across all 1000 resampled distributions for the 2020 dataset, the overall expected income was $43,304.67, however if we restrict our analysis to only those individuals who are Black *or* a woman (excluding White men but including Black men, White women, and Black women), we find that the expected income drops to $39,006.65: this difference of −$ 4298.01 is the expected income penalty associated with having *at least* one of the identities in question, without knowing explicitly which one the person had. To consider the intersectional synergy, we can also ask what the expected income for someone who is both Black *and* a woman is: $35,908.84. This is a $7395.81 financial penalty specific to black women and another $3097.80 penalty that goes above and beyond being either Black or a woman. If we consider the Venn Diagram in Figure 2, the shared cost of being Black or a woman is represented by the innermost, hatched, intersection, while the specific cost of being Black and a woman is represented by the outermost oval of synergy. This perspective is analogous to the definition of synergy given by Equation (Equation 9) and shows how the overall logic of the PID can be applied to other kinds of analysis relating “wholes” and “parts” but are not specifically operating on information directly. In the same way that synergistic information can be calculated as the difference between the whole and the union of the parts, the synergistic “cost” of misogynoire can be calculated as the difference between the cost of both identities considered jointly and the simpler redundant cost shared redundantly.

Formally:E[Income]=$43,304.67,E[Income|Black∨Woman]=$39,006.65,E[Income|Black∧Woman]=$35,908.84,

You can also do the same analysis considering privileged identities. Consider the logical disjunction of Whiteness and masculinity:E[Income|White∨Man]=$45,775.02,E[Income|White∧Man]=$56,212.66,

Being White *or* male comes with an additional benefit of $2,470.34, and the added benefit of being White and male above and beyond even that is $10,437.64, while being Black and woman comes with an intersectional “cost” of ≈$3000, being White and a man comes with an intersectional “reward” of over $10,000. This shows that there are multiple ways that intersecting identities can interact, which we might call “redundant intersectionality” vs. “synergistic intersectionality”. The redundant intersectionality is the set of experiences or effects shared by both identities that is common to both of them. In contrast, the synergistic intersectionality is the extra effect specific to the identities in question that goes “above and beyond” the shared experiences of both independent identities. For visualization of this, see Figure 5. Considering specific realizations of all the variables is analogous to performing a *local* PID [57,58], which provides a finer-grained perspective than the the expected values, but with the same logical structure of redundant, unique, and synergistic dependencies. Given a suitable, local redundancy function (such as Finn and Lizier’s $I_{\pm }$, Ince’s $I_{cs}$, or Makkeh et al.,’s $I_{sx}$, the local PID could be done in terms of bit rather than USD.

While the identity-specific decompositions described above can give us insights into how particular configurations of identities interact, each one only provides a small picture of the overall, intersectional dependencies. The local perspective does not tell us how race qua itself interacts with sex qua itself; it only gives us particular configurations of specific races and specific sexes. If we want to understand how race, as a concept that can be realized in many ways (White, Black, etc.), interacts with sex (which similarly can be realized in different ways), we need to abstract away from particular cases and consider the variables as objects in-and-of themselves. For example, the fact that there is a synergistic benefit accrued to White men does not automatically tell us anything about the interaction between race and sex more generally. The only way to get the full picture is via the partial information decomposition framework (which can *then* be localized if there is a particular interaction of interest that one wants to explore.)

### 3.2. Robustness of the Pipeline

We stress-tested our pipeline to assess how robust it might be to natural limitations in data collection: smaller sample sizes and noise in the data. By virtue of working with census data, we naturally have access to a larger-than-usual data set, although it was uncertain how well this kind of information-decomposition approach would work on smaller data sets. To test this, we re-ran the whole inference for the *I(Race, Sex; Income)* analysis, using 20 different-sized subsets of the original data set each time. Subset sizes were arranged logarithmically, from 250 samples to the full-sized data set (59,858 samples). For each subset, we randomly sampled individual respondents, and replicated each subset 600 times, to create a distribution for each subsample size. We found that, for very small subset sizes, there was a significant *over-estimation* of the difference between the joint mutual information and the sub of the marginals, although the estimates declined rapidly and converged towards the true value by ≈1000 samples (see Figure 6 Top Left). When looking at the redundancy and synergy directly (normalized by the joint mutual information to give a proportion of the total information), we found that all subsample sizes found strong evidence of synergies between race and sex on income. The specific values changed, as did the relative ratio between them, but this is a promising result that, in general, the identification of intersectional synergies is possible even in a comparatively smaller dataset (see Figure 6 Bottom Left).

To assess the effect of noise in the data, we took a similar approach. We re-ran our analysis, each time randomly permuting increasingly large subsets of the initial dataset. The 20 subset sizes were arranged logarithmically from 0.01% to 10%, and each subset size was re-tested 600 times. We found that that the whole-minus-sum heuristic showed a modest decrease as the randomization increased (see Figure 6 Top Right), however the percentage of the total mutual information that was redundant and synergistic remained extremely stable (see Figure 6 Bottom Right). These results collectively provide evidence that the PID analysis pipeline can be used for smaller, and noisier datasets while retaining sensitivity to the presence of higher-order statistical synergies.

### 3.3. Comparing PID and Linear Regression with Interaction Terms

As discussed in the Introduction, the most widely used methodology for assessing intersectionality in quantitative data is the use of multiplicative interaction terms in linear regression [19], however the linear interaction term fails to effectively disambiguate between different kinds of relationships between variables. To demonstrate this, we create two dummy datasets based on the empirical data (taken from the 2020 ASEC data), one of which is completely redundant, and another which is almost completely synergy-dominated, and show that the linear regressions (including interaction variables) are practically the same, despite the enforced differences in interaction structure. We argue that this indicates that regression-based approaches to intersectional data analysis have been missing important structures and relationships in the data that have clear implications for intersectional analysis. By constructing two datasets that are as distinct as possible in terms of underlying information structure, we aim to give the linear regression model the best possible chance of success: if the interaction term is, in some way, sensitive to redundant and synergistic interactions, then by constructing the largest possible difference, we are creating the strongest possible “signal” for the regression to pick up on.

We began with the relationship between Income and Health, known to be correlated in the real data. To create a triad dominated by redundancy, we created a dummy variable $D_{red}$ constructed in such a way that the information it contained about Health Status was completely redundant with Income:(14)$$D_{Red}=\left(1+Income\right)\;\mathrm{mod}\;4.$$

$D_{red}$ is functionally a copy of the Income dataset, with every value incremented by 1 (and wrapping around for values higher then the maximum income. Our joint mutual information decomposition then becomes:(15)$$I(Income,D_{Red};Health).$$

When we do the full partial information decomposition using the Williams and Beer redundancy function $I_{min}$, we find that, as expected, all of the joint mutual information is redundant (the results are functionally the same when using $I_{BROJA}$, see Appendix A). This is an example of what we might might refer to as “source redundancy”: all of the redundant information is present in correlations between the source variables. Harder et al., proposed a distinction between “source redundancy” and “mechanistic redundancy”, the later of which is redundancy that is somehow intrinsic to the computation performed by the target and can be non-zero even in cases where both sources are independent. This distinction has been further explored by Goodwell and Kumar [36] and Ince et al. [59,60]. Untangling source and mechanistic redundancy remains an open problem in the field, and future work may show how it is reflected differently in different statistical analyses.

The second dummy variable was constructed to result in a synergy-dominated decomposition and is given by:(16)$$D_{Syn}=\left(Income+Health\right)\;\mathrm{mod}\;2.$$

The operation is analogous to a generalized logical XOR operator (see [38] for reference). We decompose:(17)$$I(Income,D_{syn};Health).$$

to produce a partial information decomposition that is 94.4% synergistic, with the remaining 5.6% of the information distributed over unique and redundant partial information terms (it is impossible to create a purely synergistic relationship involving Income and Health Status, since Income already discloses information about Health Status and they are therefore not independent), while these two distributions are unlikely to be representative of real-world datasets (we anticipate that almost all real-world data will combine elements of all partial information-sharing modes to some extent), by constructing simple test cases, we can get a better understanding of how the PID and the linear regression relate to each-other. You can see the results of analyzing these two datasets into standard linear regressions with multiplicative interaction terms in Table 2 and Table 3.

Looking at these regressions, it is clear that, despite the completely different information-structures, the linear regressions with multiplicative interaction terms finds very similar, statistically significant coefficients. Even a more involved analysis such as a Shapely decomposition would be unable to distinguish the “types” of interaction, instead treating it as a lump sum. These results can be visualized in Figure 7. This is a significant finding because it shows that linear regression with multiplicative interactions terms is incapable of distinguishing between “synergistic” intersectional relationships where the joint effects of identities are greater than the sum of their parts, and “redundant” interactions between identities that are not non-additive in the way originally discussed by Crenshaw and Bailey.

These are admittedly somewhat constructed dummy variables: no combinations of features in the census dataset displayed “pure” information structures, instead showing a mixture of unique, redundant, and synergistic dependencies that made comparisons of this type difficult. Instead, these results are strong evidence that linear regression-based statistical analyses cannot provide a full account of all the dependences in a set of variables. Given either of the coefficients presented in Table 2 or Table 3, it would be impossible to infer which one came from the redundancy-dominated or synergy-dominated configuration. Furthermore, even though both distributions were constructed de novo to have only multivariate dependences (redundancy or synergy) and no unique pairwise dependencies, both regressions returned significant single-variable regression coefficients. This suggests that, despite the lack of any unique information in the constructed distributions, the regression is assigning some dependency to the marginal variables. These results have clear implications for quantitative analysis of intersectionality in data. Prior studies that report significant multiplicative interaction coefficients as an indicator of intersectional relationships may, in fact, be conflating different *kinds* of interactions: redundant and synergistic dependencies may be indistinguishable to the regression. Furthermore, apparently “unique” effects (associated with the regression coefficients of the marginal variables), may not always correspond to the unique information disclosed by that variable in an information-theoretic sense.

## 4. Discussion

In this work, we have shown how the formal frameworks of information theory writ large, and partial information decomposition specifically, can be used to address the question of identifying intersectionality in quantitative data. Using a decade of data from the US Census Annual Social and Economic Supplement, we show that synergistic, “greater-than-the-sum-of-their-parts” interactions can be identified in the interactions between identities such as race, sex, and class on outcomes such as income and general health status and that the relative proportions of redundant, unique, and synergistic information remain stable across the decade. We furthermore show the most widely-used analyses, such as linear regression with interaction terms struggles to discriminate between synergistic and redundant modes of information sharing, which has implications for how we might model the generative dynamics that create inequalities, as well as how we might effectively intervene to correct them.

This strongly suggests that salient social dynamics and relationships are being missed by many current approaches, and that introducing complementary frameworks alongside more standard analyses may be illuminating. This work fits into the broader project first outlined by Abbott in the seminal critique *Transcending General Linear Reality* [61]. Abbott argues that over-reliance on general linear models has influenced how researchers think about the world: that the map (consisting largely of linear relationships between interacting entities, assumptions about single effects being generated by single causes, etc.) is confused for the territory (the real world, which is highly non-linear and admits complex, potentially higher-order causal interactions). The results presented here show that the standard practices that make the assumptions Abbott describes may be missing potentially important relationships in empirical data: the existence of intersectional synergies are direct evidence of higher-order interactions between attributes that are both irreducible and, by virtue of the use of the effective information, suggestive of an effective relationship. While the evidence presented here indicates that linear interaction effects struggle to disambiguate redundant and synergistic higher-order interactions (as well as potentially presenting spurious unique information-sharing relationships), we should note that this is not the only way intersectionalities may be tested for in data using linear models. In addition to the above-mentioned Shapely decomposition, other analyses such as analysis of suppressor variables [62,63] may reveal higher-order dependencies. A rigorous, mathematical treatment linking PID and suppressor variables or Shapely decompositions may reveal previously un-recognized links between partial information decomposition and other methods.

Beyond technical and methodological advances, we feel that the conceptual distinction between “redundant” and “synergistic” interactions may be of theoretical interest. For example, some authors have suggested that intersectionality as a framework balkanizes individuals into small groups with limited shared solidarity. This concern is exemplified by by Naomi Zack, who writes:

“…as a theory of women’s identity, intersectionality is not inclusive insofar as members of specific intersections of race and class can create only their own feminisms” [64].

Zack goes on to argue:

“These ongoing segregations make it impossible for women to unite politically and they have not ended exclusion and discrimination among women, especially in the academy” [64].

Without weighing into political dimension of Zack’s argument, we claim that the partial information decomposition shows that “exclusive” intersections and “shared oppression” can co-exist in society. Continuing the example of misogynoire as a relevant case, the partial information results suggests that a given individual who is both Black and a woman will simultaneously experience: synergistic costs associated with pure misogynoire (corresponding to the synergistic part of the mutual information), “generic” anti-black racism and misogyny (corresponding to the unique part of the mutual information), as well as the “shared” costs experienced by all those who are either Black or a woman (including Black men and White women).

Different distributions of redundancy and synergy may reflect distinct generative dynamics and may have implications for our understanding of how inequalities emerge. For example, the synergy specific to Black women (“misogynoire” [3]) represents what is generally theorized as “intersectionality” and reflects a particular configuration of social factors that single out Black women in particular (such as racist stereotypes, the particular intersection of medical racism and sexism, etc.). In contrast, the “shared cost” borne by everyone who is Black or a woman (including Black men and White women) likely reflects a more general societal bias that the White man is the “default” American, and is not specific to any marginalized identity. Simply knowing that someone is Black or a woman (i.e., not White and male) is enough to prompt a downward revision of expected income. Untangling these different social dynamics may help elucidate the mechanisms by which different inequalities emerge, and inform how to most effectively rectify them.

Different information structures may also inform on the effectiveness of interventions: if the relationship between two identities and a target variable is unique, then only one variable likely needs to be intervened on, while if it is synergistic, then *both* predictor variables must be accounted for, as the effect is in the joint interaction that apparently drives the outcome. For example, if there is a cost to health uniquely disclosed by income (or financial class), then correcting that disparity should (ideally) require intervening just on financial status. In contrast, if the health disparity is synergistically predicted by *both* class and race (with little to no unique component), then intervening on individual financial situations may not work, as it is the joint-state of both that drives the disparity. If we consider, as another example, the result relating sex and race to income, the fact that there is a large synergistic effect, with little unique information disclosed by sex or race individually indicates that an attempt to improve income inequality will certainly have to wrangle with both identities simultaneously: a sex-only or race-only approach fails to account for the higher-order structures that generate the inequality.

Both of these claims are admittedly speculative at present, but stand as open questions of potential significance for future research on intersectionality.

The same analysis can be done with regards to the rewards of privilege. An interesting question for further research would be how privileged and marginalized identities interact: for conceptual simplicity we have focused on the two cases where identities “stack” in the same direction (Blackness and womanhood are both generally marginalized, while Whiteness and masculinity are generally privileged), although the synergies that may emerge when privileged and marginalized types of identities co-exist could yield interesting, novel insights. These complex interactions have been previously noted, and sometimes lead to apparently paradoxical or hard-to-understand patterns. For example, the Latino/Hispanic Health Paradox [65,66] describes the counter-intuitive finding that, despite facing an array of environmental, social, and economic challenges, Latino-Americans tend to have better health outcomes and live longer than the average American. Conversely, the finding that so-called “deaths of despair” are primarily (although not uniquely) associated with White men [67,68] suggests a complex interaction between ostensibly privileged identities and adverse outcomes such as suicide and drug addiction. By providing an analytical framework that allows for more fine-grained decomposition of interactions then is usually done, the PID framework may be able to help untangle some of these complexities.

We should stress that the framework presented here is not intended to either “prove”, or “disprove”, the existence of intersectionality as a theory. Instead, it is a statistical framework that can be used to identify irreducible intersectional synergies in large data sets. We hope that this may inform evidence-based discussion around social issues. The failure to find a particular synergistic relationship in a particular data set should *not* necessarily be grounds to claim that a particular intersection is “unreal” or “unimportant” given the restrictions inherent in working with limited data sets and the limitations of data collection. By the same token, the identification of an unexpected statistical synergy should be addressed critically and assessed in the context of previous scholarship.

Despite its utility, information theory comes with some particular limitations that must be addressed. The first is that the amount of data required for a reliable inference is much larger than what is required for a linear regression. This is particularly pressing for higher-order analyses that require reliable estimation of high-dimensional joint-probability spaces. For studies that rely on “big data” sets (e.g., Census data, data gathered from social media, etc.), this is unlikely to present a problem, although smaller-scale survey-based studies of local populations may not generate sufficient data, in which case linear methods may be preferable. Our stress-testing of our pipeline can allay those concerns somewhat, although there are known lower-bounds on the ability to reliably infer probability distributions from finite datasets [20]. Another pressing issues is that the number of distinct information-sharing modes grows super-exponentially with the number of sources: for example, assessing the intersections of 5 distinct identities (e.g., race, sex/gender, class, sexuality, ability) on a single outcome (e.g., life-expectancy) would require computing 7,828,352 distinct values (many of which would be very hard to interpret higher-order terms such as the information about life expectancy disclosed by the joint state of race and sex/gender or class or the joint state of sexuality and ability). Given this difficulty, other frameworks for information decomposition may be worth exploring, such as the “synergy-first” proposal given by Quax et al. [69] or Rosas et al. [70].

Given these limitations, as well as the undeniable successes attributable to linear regression models, we do not suggest that regressions be tossed aside in favour of the PID exclusively, but rather that these different perspectives can be combined. Both frameworks have strengths and weaknesses, and we feel that there will be many contexts in which both together will provide insights that may not be accessible to either one alone. One possible way that the two methods could inform each-other is by using linear regression as a kind of filter. Given the tremendous number of factors that can be socially relevant to a single outcome, a complete PID analysis of every possible interaction would be unwieldy at best, and impossible at worst. Linear regression-based approaches, which are much more efficient to run, may be useful for identifing a restricted set of informative sources, which can then be decomposed using PID once an interesting interaction has been identified and more fine-grained analysis is required.

The relative simplicity of applying the partial information decomposition to the effective information opens the doors to a range of future research directions. For instance, it may be worthwhile to revisit previous empirical studies of intersectionality (e.g., those discussed in [19] to disambiguate which interactions are synergistic in nature, vs. which ones are redundant in nature. It also suggests that researchers doing a multiplicative-interaction based analysis of intersectional relationships may consider supplementing their analysis with a PID-based framework to explicitly untangle redundant and synergistic components. Alternately, being able to untangle what features are redundantly predictive of an outcome, vs. which are synergistic can provide deeper insights into the generative dynamics of social relationships. Finally, recent work on generalizing the PID framework to accommodate multiple targets [71] can also expand the space of accessible interactions and further untangle complex dependencies between identities and outcomes (for example, how do race and sex joint inform on income and health together). We anticipate that, as work on PID is further developed, a large space of hitherto-unrecognized, higher-order interactions (both redundant and synergistic) will come into view and provide a rich ground on which to build a deeper understanding of complex interactions in society and health.

## 5. Conclusions

Despite the extensive and valuable work that has been done integrating intersectional frameworks in data analysis and social science research, the most commonly-used methods (e.g., linear regression with multiplicative interaction terms) fail to capture the range of ways that different identities can interact and predict outcomes. Information theory provides an appealing alternative framework that allows users to identify different modes of interactions of intersecting identifies, while simultaneously shedding the requirements imposed by assumptions of linearity. These modes, which we called “redundant intersections” and “synergistic intersections” correspond to shared costs/benefits of multiple interacting identifies and the exclusive costs/benefits, respectively. Examples can be seen in the redundant and synergistic interactions between race and sex considered jointly, allowing the identification of a strong synergistic cost of misogynoire, as well as a shared common cost between Blackness and womanhood. We hope that the framework detailed here will enable novel and insightful work on intersectionality in empirical data.

## Figures and Tables

**Figure 1 entropy-24-01387-f001:**
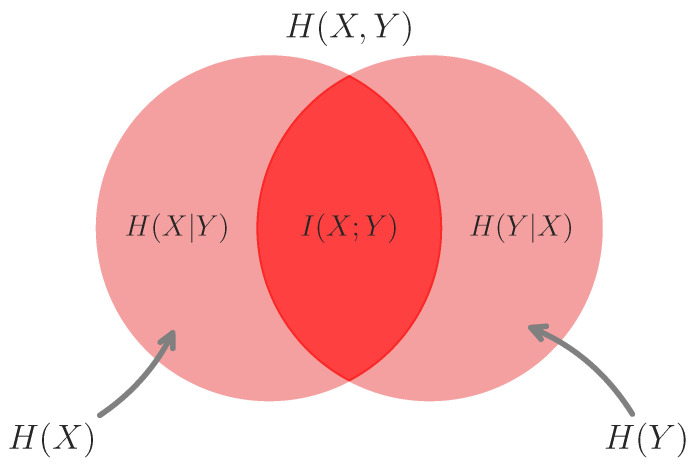
**Mutual information as the intersection of marginal entropies.** Mutual information can be understood as the intersection the entropies of two correlated variables. This plot highlights the intuition that I(X;Y)=H(X)−H(X|Y)=H(Y)−H(Y|X). Venn diagrams made using the matplotlib-venn package.

**Figure 2 entropy-24-01387-f002:**
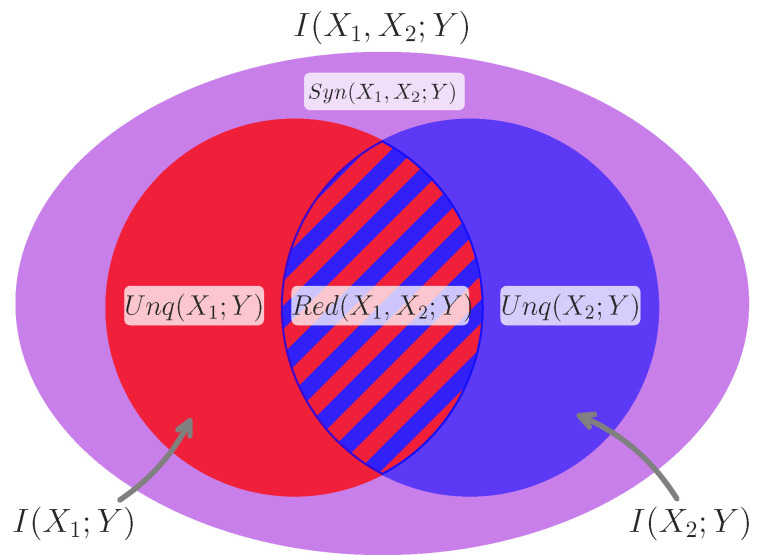
**Partial information decomposition for two predictor variables and a single predicted variable.** A Venn Diagram showing how the various components of partial information (redundant, unique, and synergistic) are related to the joint and marginal mutual information terms for two source variables $X_{1}$ and $X_{2}$, and a target variable *Y*. The two circles correspond to the mutual information between each source and the target, while the large ellipse gives the joint mutual information between both sources and the target. Notice that the marginal mutual informations overlap, each one counting the redundant (shared) information towards it’s own marginal mutual information. We can also see that $I\left(X_{1},X_{2};Y\right)>I\left(X_{1};Y\right)\cup I\left(X_{2};Y\right)$: the difference is the synergistic information which cannot be resolved to either marginal mutual information.

**Figure 3 entropy-24-01387-f003:**
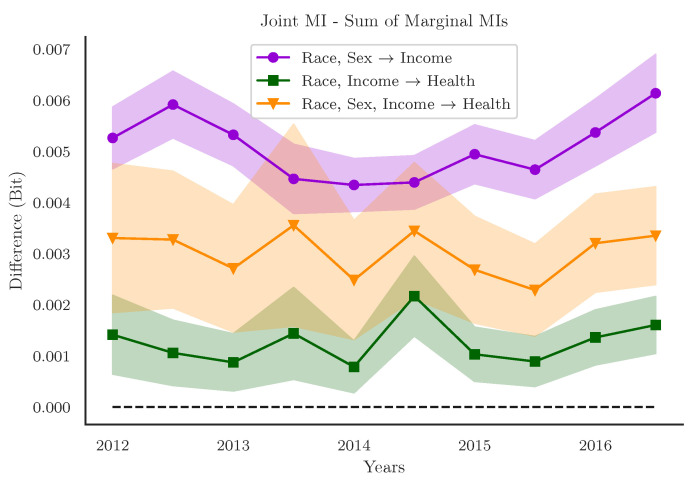
**The difference between the joint and the sum of the marginal mutual informations for three relationships.** Across all ten years, for all three relationships assessed, the difference between the joint and the sum of the marginal mutual informations was consistently greater than 0, indicating that the joint state of all identity groups considered together disclosed more information about the outcomes (income, health status) than all the identities considered independently. The error-bars indicate a range of ±1 standard deviation.

**Figure 4 entropy-24-01387-f004:**
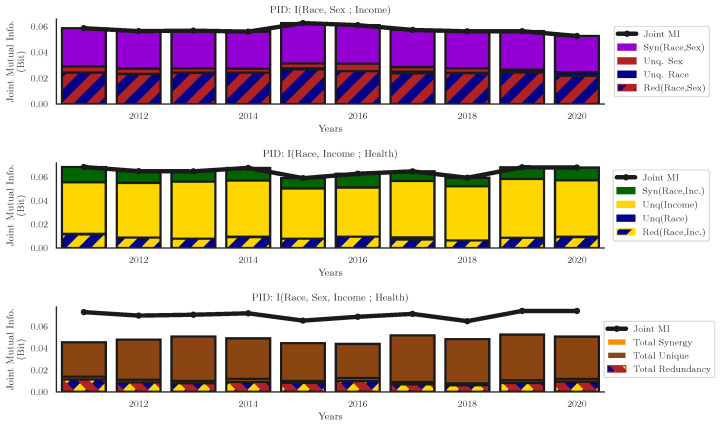
**Partial information decomposition of*****I(Race, Sex; Income)*****,*****I(Race, Income; Health Status)*****and*****I(Race, Sex, Income; Health Status)*****. Top:** The PID for race and sex on income. The information about income disclosed by race and sex is almost entirely either redundant or synergistic, race or sex individually disclose very little unique information. **Middle:** PID for race and income on health status. In contrast to the effects of race and sex, almost all information about health status is disclosed by income, although there is a non-trivial amount of both shared and synergistic information. This shows that interacting identities can have markedly different structures only revealed by information decomposition. **Bottom:** The PID for the relationship between race, sex, and income on health status. Due to the large number of distinct information-sharing modes, we aggregated all purely redundant terms, all purely unique terms, and all purely synergistic terms. The small difference between this total and the complete joint mutual information corresponds to exotic, higher-order interactions not reported here. We can see that, for all three relationships, the degree of informativeness remains remarkably constant over the decade, and that the overall degree of synergy, redundancy, and unique information is similar consistent.

**Figure 5 entropy-24-01387-f005:**
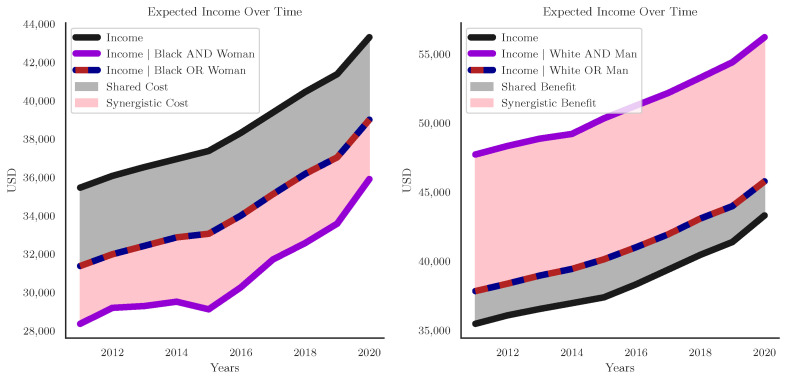
**The difference between the shared cost of marginalization, and the extra cost of synergistic intersectionality. Left:** The expected income for the whole population over time, the expected income for that subset of the population that is Black or a woman, and, and the expect income for Black women. We can see that there is an extra cost incurred by being Black and a woman that is “above and beyond” the cost incurred by being Black or a woman. **Right:** the same plot, but considering Whiteness and masculinity. Once again, the joint relationship is strong different from the disjunction, only this time there is a synergistic benefit to being White and male, as opposed to a synergistic cost.

**Figure 6 entropy-24-01387-f006:**
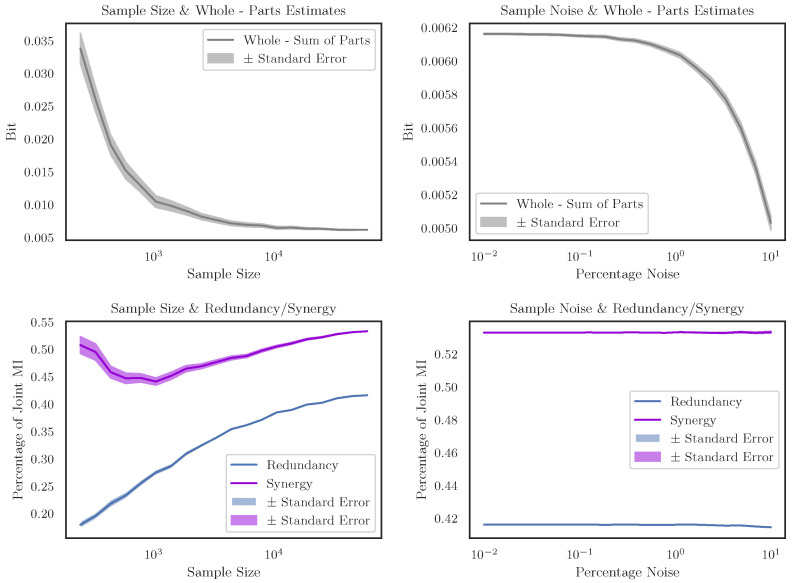
**Robustness of the PID pipeline to noise and small sample sizes.** The four plots show how the analytical pipeline described above performs when the initial dataset is shrunk, or has noise added to it. *I(Race, Sex; Income)* was used as an example data set, although comparable results can be seen for the other relationships. **Top Left** Assessing how the difference between the joint mutual information and the sum of the marginal mutual informations changes as the number of samples in the original data set is reduced. We can see that smaller samples tend to over-estimate the difference between whole - sum, although convergence happens reasonably quickly. **Top Right** Plotting the difference between the joint MI and the sum of the marginals as progressively more noise is added to the data. We can see that the result is somewhat sensitive to noise; with 10% noise in the data, the percentage change in whole-minus-sum information being ≈−18.355%. **Bottom Left** The proportion of the total MI that is redundant or synergistic as the sample size increases, while the absolute values and relative ratios of the particular atoms change, the pipeline is always able to identify the existence of synergistic relationships in the data. **Bottom Right** The proportion of MI that is redundant or synergistic as the noise increases. Despite the change in whole-minus-sum values as noise increases, the relative ratios remain largely constant. These result show that the PID inference pipeline is reasonably robust both to decreased sample sizes and noise in the data.

**Figure 7 entropy-24-01387-f007:**
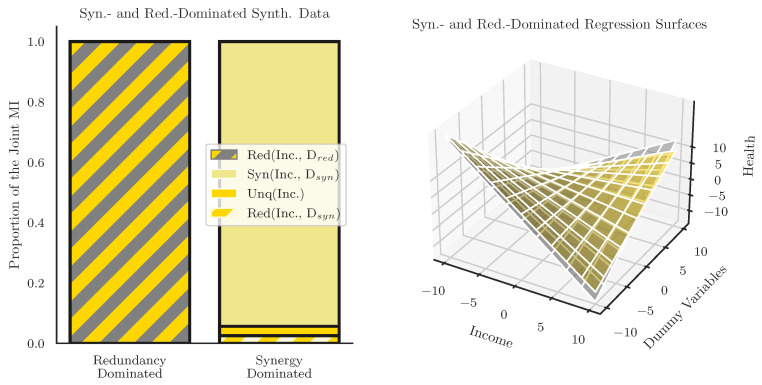
**Linear regression with interaction terms fails to differentiate between synergy- and redundancy-dominated relationships Right:** The partial information decomposition for two different dummy datasets. *I(Income, $D_{red}$; Health)* was constructed to be completely redundant: Income and $D_{red}$ disclose exactly the same information about Health. In contrast *I(Income, $D_{syn}$; Health* was constructed to be largely synergy dominated. Despite these clear differences in structure, the linear regressions with multiplicative interaction terms return essentially the same regression (see the surface plots on the right). This shows that linear regressions cannot uncover synergistic higher-order intersections of the sort that the intersectionality framework focuses on.

**Table 1 entropy-24-01387-t001:** **Income vs. Race and Sex.** Least-squares regression results for the linear interaction between Race and Sex on Income.

Variable	β	Std. Err.	T-Statistic	95% C.I.
Intercept	1.6762 *	0.007	238.521	[ 1.662, 1.690]
Race	−0.6375 *	0.020	−31.178	[−0.678, −0.597]
Sex	−0.6133 *	0.010	−62.519	[−0.632, −0.594]
Race×Sex	0.4405 *	0.027	16.140	[ 0.387, 0.494]
$R^{2}$	0.076	**No. of Observations**		59,858
**F-Statistic**	1641.0	**Prob > F**		0.00
**AIC**	$1.832\times 10^{5}$	**BIC**		$1.832\times 10^{5}$

* *p* < 10^−10^.

**Table 2 entropy-24-01387-t002:** **Redundancy Dominated Synthetic Distribution.** Least-squares regression results for the redundancy-dominated synthetic distribution. This distribution was generated using a circularly-shifted logical COPY operation.

Variable	β	Std. Err.	T-Statistic	95% C.I.
Intercept	2.8980 *	0.025	116.486	[2.849, 2.947]
Income	−0.3359 *	0.009	−39.427	[−0.353, −0.319]
$D_{Red}$	−0.3346 *	0.021	−15.794	[−0.376, −0.293]
Income $\times \;D_{Red}$	0.1336 *	0.010	13.467	[0.114, 0.153]
** $R^{2}$ **	0.069	**No. of Observations**		59,858
**F-Statistic**	1485.0	**Prob > F**		0.00
**AIC**	$1.682\times 10^{5}$	**BIC**		$1.682\times 10^{5}$

* *p* < 10^−10^.

**Table 3 entropy-24-01387-t003:** **Synergy-Dominated Synthetic Distribution.** Least-squares regression results for the synergy-dominated synthetic distribution. This distribution was constructed using the generalized logical-XOR function.

Variable	β	Std. Err.	T-Statistic	95% C.I.
Intercept	2.5841 *	0.009	291.454	[2.567, 2.601]
Income	−0.2934 *	0.005	−60.137	[−0.303, −0.284]
$D_{Syn}$	−0.1645 *	0.012	−13.554	[−0.188, −0.141]
Income $\times D_{Syn}$	0.1473 *	0.007	21.213	[0.134, 0.161]
$R^{2}$	0.071	**No. of Observations **		59,858
**F-Statistic**	1520.0	**Prob > F**		0.00
**AIC**	$1.681\times 10^{5}$	**BIC**		$1.681\times 10^{5}$

* *p* < 10^−10^.

## Data Availability

All of the data are available on the US Census website: https://www.census.gov/data/datasets/time-series/demo/cps/cps-asec.html (accessed on 9 September 2021).

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
