# Peer review of "Untangling Synergistic Effects of Intersecting Social Identities with Partial Information Decomposition"

_entropy, 2022, doi:10.3390/e24101387_

Round 1
Reviewer 1 Report
The authors introduce the usage of partial information decomposition (PID) to the analysis of data from social science. In particular, it is proposed to estimate the synergy to identify intersectional effects in social science data sets. The application of information-theoretic data analysis, especially using the PID framework is interesting and relevant. Here, the authors nicely illustrate the advantages of applying PID to empirical data over using traditional methods such as linear models with interaction terms. PID results are investigated using simulated data and linked to interpretations in terms of the original data, rather than information-theoretic units. The paper is well written, was enjoyable to read, and should be of interest to researchers both from the application as well as the methodological domain.
# Methods
In lines 250 and 251, the lattice is mentioned the first time without proper introduction. A brief explanation may be helpful for readers that are new to the topic of PID.
One approach to at least partially address the bias problem when estimating information-theoretic methods, would be to perform a permutation test on the estimated mutual informations,
I(Race, Sex; Income), I(Race, Income; Health), I(Race, Sex, Income; Health), by comparing the original estimates against a Null distribution representing the Null-hypothesis of no mutual information between the variables. Such a Null distribution may be generated by repeatedly permuting the target variable and estimating the MI using the permuted target data.
# Results
ll. 374: Linear regression is not generally insensitive to synergies and redundancies, is it? In my understanding, linear synergistic effects may also be identified in linear models and are known as suppressor effects (e.g., Friedman & Wall, 2005, The American Statistician, 59(2)). Assuming a linear model with two independent variables, a synergistic contribution may be identified by comparing the variance explained when including both predictor variables (without an interaction term), to the sum of variance explained by two models including only one of the variables each. If the sum of variance explained by the two one-predictor models is lower than the variance explained by the full two-predictor model, there exists a synergistic effect in the data. Conversely, if the summed explained variance is higher than the variance explained by the two-predictor model, there is a redundant effect. Of course, this is only an approximation to a full disentanglement of variable contributions and it does not allow to differentiate between simultaneously present synergistic and redundant effects. Also, only linear interactions are detected, while higher-order dependencies are missed. However, in my opinion, this relationship between linear models and redundant/synergistic effects is missing from the discussion of insights obtainable from PID compared to those from more traditional approaches. The question is, whether comparing synergy and redundancy to the interaction term is the correct comparison to make, from a purely statistical/methodological point of view.
Conceptually, one may even critique that using an interaction term to investigating intersectional contributions is considered the gold standard. Rather, suppressor effects between marginal variables may be the more appropriate operationalization of intersectional contributions or should at least be considered. The interaction term introduces an interaction of different order, but already a linear model with two additive terms may lead to interactions, which can be investigated by comparing the explained variance of the full to the summed variance explained by the reduced/marginal models (comprising only subsets of the predictor variables). In Section 3.3, it may be interesting to extend the analysis by comparing the explained variance of the two-predictor models to the summed explained variance of one-predictor models (each without the interaction term). This may even be an additional point to consider in the discussion of the shortcomings of linear models and interaction terms as tools to study intersectional effects.
Furthermore, when interpreting the results in Section 3.3, one could add a comment on the difference of redundancy/synergy introduced due to properties of the data or due to properties of the mechanism linking the input variables to the considered output variable (also termed source versus mechanistic redundancy/synergy e.g., Harder, Phys Rev E, 87(1)).
Minor changes:
Throughout the manuscript, the abbreviation PID and "partial information decomposition" are used alternately.
l.82: blank missing after reference [20]
l.90: period missing after "etc"
l.133: two closing brackets
ll.147: Is this sentence intended to be in italics as well?
l.180: "sane" should be "same"
l.208, l.233: period should be outside the quotation marks?
l.244: Closing brackets missing from the unique information
l.314: duplicate "are are"
l.394: ", There" change to ". There"
l.680: change "tossed side" to "tossed aside"
Author Response
You can see the response to the reviewers in the attached document.

Reviewer 2 Report
The authors of this paper show that the effects of intersectional identities can be statistically observed in empirical data using information theory, particularly the partial information decomposition framework. In addition, the authors show that, when considering the predictive relationship between various identity categories such as race and sex, on outcomes such as income, health and wellness, robust statistical synergies appear.
This is a very interesting paper which is quite well written and whose subject is very well presented. The reviewer believes that the paper can be accepted for publication. The only minor suggestion is to include a paragraph in the introduction starting with "In this paper..." describing the contributions of this work.
Author Response
You can see our response to the reviewer in the attached PDF.

Round 2
Reviewer 1 Report
I have no further suggestions and propose to publish the manuscript as is. I thank the authors for their responses and careful revision of the paper.